# Current Status and Future Perspectives of Liquid Biopsy in Small Cell Lung Cancer

**DOI:** 10.3390/biomedicines9010048

**Published:** 2021-01-07

**Authors:** Patricia Mondelo-Macía, Jorge García-González, Luis León-Mateos, Adrián Castillo-García, Rafael López-López, Laura Muinelo-Romay, Roberto Díaz-Peña

**Affiliations:** 1Liquid Biopsy Analysis Unit, Oncomet, Health Research Institute of Santiago (IDIS), 15706 Santiago de Compostela, Spain; patricia.mondelo.macia@sergas.es (P.M.-M.); lmuirom@gmail.com (L.M.-R.); 2Department of Medical Oncology, Complexo Hospitalario Universitario de Santiago de Compostela (SERGAS), 15706 Santiago de Compostela, Spain; jorge.j.garcia.gonzalez@gmail.com (J.G.-G.); luis.angel.leon.mateos@sergas.es (L.L.-M.); rafa.lopez.lopez@gmail.com (R.L.-L.); 3Translational Medical Oncology (Oncomet), Health Research Institute of Santiago (IDIS), 15706 Santiago de Compostela, Spain; 4Centro de Investigación Biomédica en Red de Cáncer (CIBERONC), 28029 Madrid, Spain; 5Fissac-Physiology, Health and Physical Activity, 28015 Madrid, Spain; adrian@fissac.com

**Keywords:** lung cancer, SCLC, liquid biopsy, cfDNA, CTC

## Abstract

Approximately 19% of all cancer-related deaths are due to lung cancer, which is the leading cause of mortality worldwide. Small cell lung cancer (SCLC) affects approximately 15% of patients diagnosed with lung cancer. SCLC is characterized by aggressiveness; the majority of SCLC patients present with metastatic disease, and less than 5% of patients are alive at 5 years. The gold standard of SCLC treatment is platinum and etoposide-based chemotherapy; however, its effects are short. In recent years, treatment for SCLC has changed; new drugs have been approved, and new biomarkers are needed for treatment selection. Liquid biopsy is a non-invasive, rapid, repeated and alternative tool to the traditional tumor biopsy that could allow the most personalized medicine into the management of SCLC patients. Circulating tumor cells (CTCs) and cell-free DNA (cfDNA) are the most commonly used liquid biopsy biomarkers. Some studies have reported the prognostic factors of CTCs and cfDNA in SCLC patients, independent of the stage. In this review, we summarize the recent SCLC studies of CTCs, cfDNA and other liquid biopsy biomarkers, and we discuss the future utility of liquid biopsy in the clinical management of SCLC.

## 1. Introduction

Small cell lung cancer (SCLC) accounts for approximately 15% of all lung cancers diagnosed worldwide and up to 25% of lung cancer deaths [1]. The majority of patients with SCLC present with metastatic disease, and the prognosis is poor, with less than 5% of patients alive at 5 years. SCLC diagnosis is performed by morphological assessment of a tissue biopsy or cytology sample. Radiological staging with the conventional TNM criteria is recommended, but according to the Veterans Administration of Lung Study Group staging, SCLC is classified into two stages: limited disease (LD-SCLC), when it is confined to a hemithorax, where curative treatment with radiochemotherapy is feasible; or extensive disease (ED-SCLC), defined as the presence of metastatic disease outside the hemithorax at first diagnosis [2].

Liquid biopsy, principally circulating tumor cells (CTCs) and cell-free DNA (cfDNA), has the potential to improve the management of SCLC at different levels: screening, minimal residual disease detection, monitoring systemic treatment initiation and response and determining the development of resistance. These techniques generate promising data to better understand SCLC biology and guide patient treatment [3]. This review will focus on the current and future clinical utility of liquid biopsies, which represent a key factor in improving outcomes for patients with SCLC.

## 2. Clinical Management of SCLC

### 2.1. Small Cell Lung Cancer Treatment Landscape

Systemic treatment of SCLC patients is based on the combination of platinum and etoposide-based chemotherapy, and recently, the use of immunotherapy has also been incorporated in patients with metastatic disease [4,5,6,7]. The goal of LD-SCLC therapy is to cure. Surgery is an option to treat patients who are diagnosed with tumors smaller than 3 cm in size and without hilar lymph node metastasis [7]. However, in most patients, the standard treatment is the combination of platinum plus etoposide chemotherapy administered concurrently with radiotherapy. Unfortunately, the prognosis of these patients is poor; specifically, the median overall survival (OS) of patients with LD-SCLC ranges from 23 to 30 months, with an estimated 2-year survival rate ranging from 47 to 56% [4,5]. Radiation therapy is also an important part in the treatment of every stage of SCLC, in curative as well as in palliative therapy, and it is changing in recent years [8]. For this reason, its role remains somewhat controversial [9,10]. There is consensus that thoracic radiation should be considered in patients with residual disease after chemotherapy, with a range of doses being appropriate. Moreover, it must also be considered after chemo-immunotherapy in patients with residual disease [9].

Regarding their response to first-line treatments, SCLC patients are classified as chemorefractory or chemosensitive, and this trait is a key factor in guiding the choice of second-line therapy. Chemorefractory disease is defined as the occurrence of progression during first-line therapy or within 90 days of its completion. If the time to progression after chemotherapy is greater than 90 days, the disease is defined as chemosensitive. This classification is a prognostic factor for early recurrence and reduced response to subsequent cytotoxic therapies; however, it might not be relevant to the clinical development of novel therapies [11].

Unfortunately, the majority of SCLC patients progress on treatment, and the choice of the second-line treatment is topotecan or cyclophosphamide, doxorubicin and vincristine (CAV), with poor results [12]. However, recently, new drugs have been approved in this setting. For instance, lurbinectedin has been approved by the US Food and Drug Administration (FDA) for salvage treatment of SCLC that has relapsed from platinum compound-based, first-line chemotherapy [13]. Currently, the efficacy of the lurbinectedin plus doxorubicin combination is being investigated in the randomized phase 3 trial ATLANTIS (NCT02566993) [14]. However, this study appears not to meet the pre-specified criteria of significance for the primary endpoint of OS in the intent-to-treat (ITT) population, comparing lurbinectedin in combination with doxorubicin to the control arm (https://pharmamar.com/wp-content/uploads/2020/12/PR_PhM_and_Jazz_anounce_results_ATLANTIS_DEF.pdf, last accessed on 21 December 2020). Nivolumab and pembrolizumab have also been approved by the FDA for refractory SCLC based on the results of the CheckMate-032 (NCT01928394) and Keynote-158 (NCT02628067) clinical trials, respectively [15,16]. However, in the Checkmate-331 trial (NCT02481830), nivolumab failed to demonstrate an improvement in OS versus standard treatment in patients with relapsed or refractory SCLC treated with a platinum-based line of chemotherapy [17].

On the other hand, first-line treatment in patients with ED-SCLC is undergoing great changes, especially in the field of immunotherapy. For example, blocking the programmed death 1 receptor (PD-1) or its ligand (PD-L1) pathway has been shown to improve the prognosis of patients with ED-SCLC [18,19]. The Impower 133 (NCT02763579) and CASPIAN phase III trials (NCT03043872) demonstrated that adding atezolizumab or durvalumab to platinum plus etoposide chemotherapy improves OS and progression-free survival (PFS) compared to chemotherapy alone [20,21]. More recently, the KEYNOTE-604 (NCT03066778) study reported that combining pembrolizumab with platinum and etoposide improves PFS but does not significantly improve OS in patients with ED-SCLC [22].

### 2.2. New Therapies

In another type of tumor, non-small cell lung adenocarcinoma, growing knowledge of its molecular characteristics, especially driver mutations, has been accompanied by the development of targeted drugs that have significantly improved survival. By contrast, the identification of new therapeutic targets in SCLC has been challenging, partly because driver mutations are primarily loss of function (genes *RB1* and *TP53*) or currently difficult to target [23,24].

Recent advances have made it possible to deepen the molecular knowledge of SCLC, which has allowed us to distinguish between four different subtypes characterized by distinct gene expression profiles. These subtypes are the neuroendocrine subtypes SCLC-A (*ASCL1*-positive) and SCLC-N (*NEUROD1*-positive) and the non-neuroendocrine subtypes SCLC-P (*POU2F3*-positive) and SCLC-Y (*YAP1*-positive) [25]. Previous work has demonstrated activity of CDK4/6 inhibitors in *RB1* wild-type SCLC cell lines [26,27]. Based on these results, a clinical trial is testing abemaciclib, a CDK4/6 inhibitor, as a single agent in ED-SCLC patients with wild-type *RB1*, with platinum refractory disease (ClinicalTrials.gov Identifier NCT04010357).

Moreover, genomic analyses have determined that SCLC tumors present extensive copy number alterations (CNAs) and high mutation rates, although SCLCs are not characterized by homologous recombination (HR) deficiency or mutations in *BRCA1/2* [28,29]. The development of new drugs for SCLC patients should be based on a better understanding of the nature of these subtypes, characterizing their differences, similarities and what determines their growth, patient survival rates and tendency to metastasize. In this sense, poly [ADP ribose] polymerase (PARP), delta-like canonical Notch ligand 3 (DLL3) in SCLC-A subtypes and Aurora kinase in SCLC-A constitute promising drug targets [30,31].

Veliparib is an oral PARP inhibitor that potentiated standard chemotherapy against SCLC in preclinical studies. The addition of veliparib to front-line cisplatin and etoposide chemotherapy has shown a potential benefit in PFS in ED-SCLC [32]. In relapsed or refractory SCLC patients, veliparib plus temozolomide failed to find improvement in PFS compared to placebo plus temozolomide [33]. However, in patients with high *SLFN11* expression, veliparib significantly prolonged PFS and OS; thus, SLFN11 has been postulated as a new drug target [33,34]. Although it has not yet found an application in daily clinical practice, the usefulness of SLFN11 expression is being explored to select patients who may benefit from adding veliparib to maintenance treatment with atezolizumab in stage IV SCLC patients (ClinicalTrials.gov Identifier NCT04334941).

Rovalpituzumab tesirine (Rova-T) is an antibody–drug conjugate designed to specifically target delta-like ligand 3 (DLL3). Initially, in a human phase I trial, Rova-T showed a promising signal of activity [35]. Unfortunately, Rova-T has not been superior to second-line treatment in two randomized trials, and its development is compromised in this setting. Alisertib is an investigational, oral, selective inhibitor of Aurora kinase A. In patients with refractory or recurrent SCLC, alisertib has been shown to be active both in monotherapy and in combination with paclitaxel [36,37].

All of these therapeutic approaches have opened new horizons and hopes for improving SCLC outcomes. However, more research is still required to reach a better understanding of the molecular mechanisms that can be used to find new predictive biomarkers to apply the most convenient therapeutic strategy for each patient, and the field of liquid biopsy would be crucial for this objective.

## 3. Liquid Biopsy as a Clinical Tool in SCLC

Tumor molecular profiling is essential for optimizing treatment in clinical practice, and tissue biopsy or cytology samples remain the gold standard. However, a tissue biopsy requires invasive procedures and is not always possible or repeatable (Figure 1). In lung cancer, fine-needle aspirates or biopsy samples are often of poor quality or quantity, and up to 25% of biopsies fail to obtain enough tissue for assessment [38], justifying the need to explore new diagnostic tools. In SCLC, due to their intratumoral and intertumoral heterogeneity, tumor samples obtained from a single biopsy may not capture the complete genomic profile of tumors, considering that the biopsy is a partial photograph of the tumor at a specific time. Additionally, the genomic landscape of SCLC can change during the course of therapy, and the follow-up of this dynamic process with repeated sampling is essential, which is difficult to address based on solid samples. Liquid biopsy has emerged as a rapid and noninvasive alternative to obtain a more complete image of the disease and to monitor its evolution over time and as a consequence of treatment pressure [39,40]. Indeed, recently, the possibility of early detection of cancer using liquid biopsy has been reported in diverse tumor types [41,42], although in SCLC, this application has been infrequently explored.

Overall, liquid biopsy is used to refer to analytes from various biological fluids, such as blood or other accessible fluids: ascites, pleural effusions, saliva or urine. The term primarily refers to circulating tumor cells (CTCs), cell-free DNA (cfDNA), circulating cell-free RNA (cfRNA) and circulating extracellular vehicles (EVs), among others [43] (Figure 1). Clinical applications of these circulating biomarkers include the early detection of cancer, tumor recurrence, therapy monitoring, detection of drug resistance and predictive response [38,44]. Although the application of liquid biopsy is infrequent in SCLC, promising advances have been achieved in recent years. In this review, we summarize the current state of knowledge on CTCs and cfDNA assays in SCLC. We also highlight potential future research directions in liquid biopsy-based approaches.

### 3.1. Cell-Free DNA

Cell-free DNA (cfDNA) was first reported in healthy and diseased individuals by Mandel and Métais in 1948 [45]. Later, Leon et al. described cfDNA for the first time in the field of oncology, reporting cfDNA levels higher in cancer patients than in healthy individuals [46]. cfDNA comprises extracellular DNA molecules (double-stranded DNA and mitochondrial DNA) released into the blood and other fluids through different mechanisms, including apoptosis, necrosis, senescence and active secretions [47,48] (Figure 2A). The short half-life of cfDNA varies from several minutes to 1–2 h, enabling real-time monitoring of molecular biomarkers for response or relapse and serving as a perfect biomarker to monitor cancer evolution [47].

During the last decade, a type of cfDNA, the tumor-derived fraction of cfDNA, or circulating tumor DNA (ctDNA), specifically present in cancer patients, has received much attention in the field of oncology (Figure 2A). The circulating tumor DNA concentration varies from 0.01% to 1% of total cfDNA in early-stage disease to over 40–90% for late-stage disease [49,50]. It tends to be more fragmented than cfDNA, with sizes ranging from 90 to 150 base pairs. Circulating tumor DNA has value in early cancer detection and can be used to determine the tissue of origin and prognosis; to monitor response and assess potential resistance to the treatments; and to detect minimal residual disease [51] (Figure 2B). Of note, a desired cfDNA assay should: (1) quantify ctDNA in order to monitor the presence and extend the disease and (2) identify ctDNA alterations that can help to make therapeutic decisions. In fact, ctDNA detection technologies could lead to personalization of therapy based on the identification of somatic alterations present in tumors and their changes during therapy administration, which have been associated with clinical outcomes in several cancer types [52,53,54,55,56].

#### 3.1.1. Circulating Tumor DNA Detection Technologies

The low concentration of ctDNA into total cfDNA involves a challenge for detecting genetic alterations (point mutations, CNAs or small indels), particularly at the early stages of tumor development. Two principal approaches have been developed to solve this challenge. First, the detection of single or a low number of point mutations using highly sensitive and specific techniques has a rather fast and cost-effective rate [57]. These techniques include assays based on qPCR, such as the Cobas EGFR mutations Test v2, which was approved in 2016 by the FDA and the European Medicines Agency (EMA) in NSCLC patients [58], and the Idylla platform. Additionally, assays based on digital PCR (BEAMing and droplet digital PCR (ddPCR)) can detect specific known mutations, showing high concordance with results obtained in tumor tissue [59,60,61], with high sensitivity (0.001–0.005%) [62]. However, in these approaches, previous information about the tumor type and the common mutations is needed. The second type of approach is focused on a genome-wide analysis of CNAs or point mutations through next-generation sequencing (NGS) strategies. This genome-wide characterization allows a more complete and patient-specific genotyping to assess tumor heterogeneity and to follow the clonal evolution across the treatment. The principal limitations of NGS-based strategies are the high cfDNA input requirement and general low sensitivity (5–10%) [51,57]. Fortunately, the technologies are improving very fast, and there are options of comprehensive targeted NGS panels able to detect genetic alterations present in lower frequencies <1% [63]. More recently, a third strategy based on the combined analysis of DNA methylation and fragmentation with targeted or wide-genome sequencing to improve ctDNA detection in early tumor stages has been described [64,65]. It has been well reported that smoking history, environment and age affect DNA methylation status [66,67]. Therefore, methylation analyses of cfDNA in cancer, and specially in SCLC patients, where the rate of smokers is high, should be performed with caution.

#### 3.1.2. Cell-Free DNA Studies in SCLC

Due to the rapid growth and highly metastatic capacity of SCLC tumors, as well as the high levels of cfDNA in SCLC patients, cfDNA analyses should allow for better characterization of the tumor. However, few studies have analyzed the role of cfDNA in SCLC (Table 1). *TP53* and *RB1* alterations play an essential role in SCLC tumorigenesis [28] and are also present at relapse, representing valuable targets to monitor ctDNA levels. Thus, Fernandez-Cuesta et al. studied the possibility of detecting *TP53* mutations in SCLC patients using cfDNA samples [49]. Their results show the potential to detect *TP53* mutations in cfDNA of 49% of patients (with different tumor stages) and, more importantly, in the 35.7% of early-stage patients, demonstrating the potential of cfDNA as a promising tool for the early detection of SCLC patients with *TP53* mutations. However, they also found the presence of *TP53* mutations in healthy controls (11.1%). In fact, the appearance of germline mutations due to clonal hematopoiesis constitutes an aspect to take into account for the development of ctDNA screening tests. This finding has led to the recommendation to include the blood cell fraction in NGS studies, although the percentage of variants associated with the phenomenon of clonal hematopoiesis is normally low [68]. Based on an NGS strategy, Almodovar et al. analyzed serial plasma samples of 11 LD-SCLC and 16 ED-SCLC patients using a custom panel of the most frequently mutated genes in SCLC tumors, including *TP53* and *RB1* [69]. Somatic variations were found in 85% of the patients, with *TP53* and *RB1* being the most common altered genes, in line with previous studies [70]. The results of this study also showed that ctDNA monitoring could identify disease recurrence prior to disease progression detected on imaging or in cases with ambiguous imaging findings. Moreover, increased cfDNA levels in SCLC patients were associated with worse OS [69]. To determine the subclonal architecture of SCLC and its molecular evolution during treatment, Nong et al. analyzed cfDNA samples from 22 SCLC patients before treatment onset and at different points within therapy using a panel that covers 430 genes altered in cancer [71]. They detected mutations in all patients at baseline, the most common being *TP53* mutations, observed in 91% (20/22) of patients, together with *RB1* mutations, observed in 64% (14/22) of patients. Moreover, in eight patients, both plasma and tissue samples were analyzed, showing a concordance of 94% for mutations, indicating that cfDNA sequencing is a sensitive tool to detect somatic mutations in SCLC patients. Despite this high concordance, in one patient, none of the 26 mutations detected in tumor tissue were detected in cfDNA, while two of the discordant cases became positive after increasing the sequencing depth. Importantly, a subset of mutations was exclusively detected in cfDNA in some patients. In addition, high cfDNA levels were associated with significantly worse PFS and OS. Overall, this study demonstrated a similar subclonal architecture between tissue and cfDNA, supporting the use of cfDNA for the detection of somatic mutations and studying the molecular heterogeneity of SCLC [71].

Similarly, Du et al. demonstrated the feasibility of using cfDNA for genomic profiling and prognostication in a cohort of 24 SCLC patients (11 LD-SCLC and 13 ED-SCLC) [72]. They reported the presence of CNAs in 66.7% of patients and the presence of point mutations in all patients. The most common genetic alterations occurred in *KMT2D* and *NCOR1,* while the presence of *SETBP1*, *PBRM1*, *ATM* and *ATXR* mutations was associated with patient survival. A higher mutation risk index was strongly related to poor PFS and OS. A subsequent study demonstrated that a simple cfDNA-based genome-wide copy number approach was an effective tool for monitoring patients during treatment and that targeted cfDNA sequencing identifies potential therapeutic targets in more than 50% of SCLC patients [73]. CNAs were observed in 84% of patients, with a gain in copy number for *SOX2, MYC, NFIB* and *CD274* and losses for *CNTN3*, *FHIT*, *RASSF1*, *RB1* and *KIF2A*, showing significant differences between LD-SCLC and ED-SCLC.

**Table 1 biomedicines-09-00048-t001:** Summary of cfDNA studies in SCLC.

Study	Cohort (LD-SCLC/ED-SCLC)	Method	Main Results
Morgensztern D et al., 2016 [70]	134 patients	NGS (Guardant360)	132/134 (92.3%) of patients had at least one mutation. The most common mutations were found in *TP53* (70%) and *RB1* (32%).
Fernandez-Cuesta L et al., 2016 [49]	51 (42/9) patients and 225 healthy controls	NGS (Custom panel)	25/51 (49%) patients had mutations in *TP53.* 25/225 (11.1%) controls had mutations in *TP53.*
Ou S.H.I et al., 2017 [74]	1 NSCLC patients transform to SCLC	NGS (FoundationACT)	An NSCLC with an *ALK* fusion transforms the histology into SCLC.
Almodovar K et al., 2018 [69]	27 (11/16) patients	NGS (Custom panel of 14 genes)	23/27 (85%) patients had disease-associated mutations. The most common altered genes were *TP53* (70%) and *RB1* (52%). Changes in cfDNA levels correlated with response to therapy and relapse.
Nong J et al., 2018 [71]	22 (11/11) patients	NGS (Custom panel of 430 genes)	All patients had at least one mutation. Mutations were mostly found in *TP53* (91%) and *RB1* (64%). 94% of mutations detected in tumor DNA were also detected in the paired ctDNA sample. High cfDNA levels in SCLC patients were associated with significantly worse PFS and OS.
Du M et al., 2018 [72]	24 (11/13) patients	WGS for CNA and NGS (xGen Pan-Cancer Panel)	16/24 (66.7%) of patients had CNAs. All patients had any mutation. *TP53* and *RB1* mutations were found in 4/17 patients (23.5%). A higher mutation risk index was strongly associated with poor PFS and OS.
Devarakonda S et al., 2019 [75]	564 patients	NGS (Guardant360)	94% of patients had at least one mutation or amplification. The most common mutations were found in *TP53* (72.5%) and *RB1* (18%). A higher percentage of alterations in *APC* and *AR* were observed in samples obtained at relapse.
Mohan S et al., 2020 [73]	69 (39/30) patients and 32 healthy controls	WGS for CNA and NGS (Custom panel of 110 genes)	58/62 (94%) of patients had at least one mutation. Most of the mutations were found in *TP53* (79%) and *RB1* (34%). 0/23 (0%) controls had genetic alterations. The presence of CNAs was associated with OS, being a potential prognostic factor and monitoring tool in SCLC patients.
Iams W.T et al., 2020 [76]	23 LD-SCLC patients	NGS (Custom panel of 14 genes)	Detection of ctDNA in patients with LS-SCLC after curative intent therapy predicts disease relapse and death.

Abbreviations: SCLC: small cell lung cancer; NGS: next-generation sequencing; NSCLC: non-small cell lung cancer; WGS: whole-genome sequencing; CNA: copy number alteration; cfDNA: cell-free DNA; ctDNA: circulating tumor DNA; PFS: progression-free survival; OS: overall survival LD-SCLC: limited disease SCLC.

Devarakonda et al. analyzed cfDNA from a larger cohort of SCLC patients using NGS at diagnosis and when the disease relapsed [75]. The results reported that 552 of a total of 609 (94%) patients had at least one point mutation or amplification (531/609). The most commonly altered genes were *TP53* (72.5%) and *RB1* (18%). Moreover, the authors observed a higher percentage of alterations in *APC* and *AR* in samples obtained at relapse. Recently, it has been hypothesized that the detection of ctDNA in patients with LS-SCLC after definitive therapy would predict disease relapse and death, after showing that residual ctDNA can be detected prior to radiographic relapse [76]. Overall, all of these studies demonstrate that cfDNA analyses constitute a powerful tool for SCLC diagnosis, therapy selection and monitoring that can be a key element to improve this tumor management in the close future.

### 3.2. Circulating Tumor Cells

CTCs are tumor cells originating from the primary or metastatic sites that are able to enter the circulation and disseminate to distant sites. Peripheral blood offers the possibility to analyze the presence of this circulating tumor population for cancer diagnosis and disease monitoring. The importance of CTCs for diagnostic and prognostic purposes has been well reported in different cancer types, such as metastatic breast [77,78], prostate [79], NSCLC [80] and colorectal cancer [81]. Of note, in SCLC, several studies have reported higher CTC levels in comparison to other cancer types [82], supporting the interest of this circulating population as an accessible tumor biopsy (Table 2).

The proportion of CTCs in the bloodstream is very low, approximately 1 CTC per 10^6^–10^7^ leukocytes [83]. CTCs are released from the primary tumor into the blood and have a very short half-life (1–2.4 h) [84]. The majority of these cells are rapidly cleared; however, some of them evade recognition by immune cells, survive into the bloodstream and can reach distant locations to generate metastasis [85,86,87]. These cells are characterized by a hybrid phenotype in terms of epithelial and mesenchymal markers that favor their survival in circulation [86,87,88,89].

#### 3.2.1. Methods to Isolate and Analyze CTCs

The low proportion of CTCs in the bloodstream together with the molecular heterogeneity that characterizes these cells is the principal challenge for CTC isolation and detection. Several CTC detection platforms have been developed in the past decade. Each technology isolates CTCs and focuses on differential features between CTCs and blood cells, such as protein expression, morphology, volume and biophysical properties, presenting different advantages and limitations.

These technologies can be categorized based on the method of isolation as label-dependent (affinity-based) or label-independent [90] (Figure 3A). The label-dependent methods are the most commonly used approaches and are based on the expression of cell surface markers (such as EpCAM). These methods use antibodies against membrane markers conjugated with magnetic nanoparticles or immobilized on the walls of microfluidic chips. They are also called immune-based detection methods, and the most commonly used is the CellSearch system (Menarini, Silicon Biosystem, Bologna, Italy). The CellSearch system employs anti-EpCAM-coated ferrofluid nanoparticles for the selection of EpCAM-positive CTCs. Then, after an immunostaining step, CTCs are discriminated from leukocytes based on the positive expression of cytokeratins and the absence of CD45 staining (leukocyte marker) together with morphologic criteria. This system has been used for many years as the reference technology because it is a unique system with FDA approval for clinical use in metastatic breast, prostate and colorectal cancer [81,91,92]. However, this approach has the limitation of only isolating CTCs based on EpCAM expression, ignoring other low epithelial CTCs [93]. On the other hand, label-independent approaches allow the isolation of CTCs with a low epithelial phenotype because these platforms discriminate CTCs from other cells based on physical characteristics such as size, density, deformability and electrical properties [90]. For example, there are size-based separation methods [94] (filter-based detection, microfluidic chips, centrifugal forces or inertial focusing) and direct imaging and biophysical property-based methods [90,95]. Other methods combine antigen-based capture with the advantages of microfluidics methods, such as CTC-iChip for increasing the isolation efficacy [90]. Due to the similar size of leukocytes and CTCs in SCLC patients, label-independent approaches based on the cell size are less efficient than in other tumors [96]. This carryover of leukocytes is seen in all CTC enrichment platforms, and when the objective is the enumeration of CTCs, a posterior step of labeling CTCs with specific antibodies is needed. Furthermore, after the detection of CTCs, some platforms allow the isolation of pure CTCs at the single-cell or cluster level by the use of micromanipulation or via dielectron force manipulation (DEPArray system), among other strategies [97] (Figure 3B). Analyses of individual CTCs/clusters at the DNA, RNA or protein level provide valuable information about the molecular heterogeneity of these cells and a more precise characterization of the disease [98].

#### 3.2.2. Circulating Tumor Cells in SCLC

The most commonly used platform to isolate CTCs in SCLC studies has been the CellSearch platform (Table 2), reporting a detection rate of 60.2–94% of SCLC patients with at least 1 CTC per 7.5 mL of blood before treatment [99,100,101,102,103,104,105,106,107]. Naito et al. detected ≥2 CTCs/7.5 mL in 68% of SCLC patients before treatment [108]. This percentage of positivity is clearly higher than in other tumors. Furthermore, high CTC numbers have been associated with the clinical stage and presence of liver metastases, representing a prognostic factor for survival [100,109]. In fact, numerous studies have reported differences in the number of CTCs detected between patients with LD-SCLC and ED-SCLC. Hilterman et al. reported that the levels of CTCs detected in LD-SCLC patients (median = 6 CTCs/7.5 mL of blood) were lower than those in patients with ED-SCLC (median = 63) [99]. Similar results were reported by Aggarwal et al., with a median of 1.5 CTCs/7.5 mL of blood in LD-SCLC patients and a median of 71 CTCs in ED-SCLC patients [103].

In line with these results, several studies have reported the association of CTC levels with patient outcomes. Hilterman et al. showed that the presence of ≥2 CTCs/7.5 mL of blood was the strongest prognostic factor to predict OS in a cohort of 59 patients with SCLC [99]. A meta-analysis also supported the prognostic value of CTCs in a study analyzing a total of 440 patients diagnosed with SCLC [110]. The study concluded that the presence of ≥2 CTCs/7.5 mL of blood was significantly associated with poorer OS and PFS, although the CTC isolation methods varied across the studies included in the meta-analysis. Despite these data, the prognostic and pharmacodynamic importance of CTCs has not reached a consensus regarding the optimal cut-off for clinical prognostication. Hou et al. reported that the persistence of more than 50 CTCs after one chemotherapy cycle is strongly associated with prognosis, showing clinical utility to guide treatment decisions even in clinical trials that test novel agents [100]. Due to the molecular heterogeneity of CTCs, studies using label-independent platforms have also reported correlating the presence of CTCs with prognosis [96,111,112,113,114]. Recently, Obermayr et al. showed that CTCs isolated by the microfluidic platform Parsortix can be assessed using epithelial and neuroendocrine cell lineage markers at the molecular level, showing the potential utility of label-independent platforms to detect CTCs during epithelial-to-mesenchymal transition (EMT) [112]. Moreover, RNA analyses of isolated CTCs showed that the expression of a Notch pathway delta-like 3 ligand (DDL3), the actionable target of rovalpituzumab tesirine (Rova-T), is associated with PFS and OS in SCLC patients. However, the number of CTC-positive samples using the Parsortix system was smaller than that reported using the CellSearch system in other studies [115]. In contrast, Chudziak et al. reported a good rate of detection in their cohort (n = 12/12 SCLC patients with at least 1 CTC) using the Parsortix system [96].

**Table 2 biomedicines-09-00048-t002:** Summary of studies that analyzed circulating tumor cells (CTCs) in SCLC.

Study	Cohort (LD-SCLC/ED-SCLC)	Platform	Main Results
Hou J.M et al., 2009 [109]	50 (20/30)	CellSearch with CD56 in the 4th channel	CTCs were detected in 86% of patients at baseline. Increased CTC counts at baseline and after 22 days of treatment were associated with worse OS.
Hiltermann T.J.N et al., 2012 [99]	59 (21/38)	CellSearch	CTCs were detected in 73% of patients at baseline. At baseline, the presence of <2 CTCs was associated with OS. Lower number of CTCs in LD-SCLC (median = 6) compared with ED-SCLC (median = 63). CTCs levels decreased after one cycle of chemotherapy. No changes were found after 4 cycles of treatment.
Hou J.M et al., 2012 [100]	97 (31/66)	CellSearch	CTCs were detected in 85% of patients at baseline. OS was poorer for patients with ≥50CTCs/7.5 mL at baseline.
Naito T et al., 2012 [108]	51 (27/24)	CellSearch	≥2 CTCs were detected in 68.6% of patients at baseline. Lower number of CTCs in LD (median = 1) compared with ED-SCLC (median = 9.5). Patients with ≥8 CTCs had worse survival than those with <8 CTCs.
Igawa S et al., 2014 [113]	30 (8/22)	TelomeScan	CTCs were detected in 96% of patients at baseline. ≥2 CTCs at baseline was associated with OS.
Normanno N et al., 2014 [104]	60 ED	CellSearch	CTCs were detected in 90% of patients at baseline. Reduction in CTC number >89% was associated with lower risk of death.
Huang C.H et al., 2014 [106]	26 ED	CellSearch	Non-significant association was reported between CTCs at baseline and PFS, or OS, but trended towards significance.
Cheng Y et al., 2016 [105]	86 ED	CellSearch	CTCs were detected in 87.6% of patients at baseline. ≥10 CTCs levels were associated with OS. ≥10 CTCs after the second chemotherapy cycle was associated with PFS and OS.
Chudziak J et al., 2016 [96]	12 (2/10)	Parsortix vs. CellSearch	CTCs were detected in 100% of patients at baseline using Parsortix and in 83.33% of patients using CellSearch. Parsortix system is a valuable tool for CTC enrichment that enables CTC analysis independently on surface epitopes.
Carter L et al., 2017 [11]	13	CellSearch and DEPArray	CTCs were detected in 100% of patients at baseline. Single-cell analyses of CTCs allowed classifying patients with chemosensitive disease and chemorefractory disease.
Messaritakis I et al., 2017 [101]	62 (22/40)	CellSearch	CTCs were detected in 62.1% of patients at baseline. Increased CTC count at baseline was associated with worse PFS and OS. Increased CTC count after one cycle of treatment was associated with poorer OS.
Messaritakis I et al., 2018 [102]	108 (37/71)	CellSearch	CTCs were detected in 60.2% of patients at baseline. Increased CTC count at baseline and at progressive disease was associated with worse PFS and OS, respectively.
Salgia R et al., 2017 [107]	42 ED-SCLC in clinical trial	CellSearch + CXCR4 expression	CTCs were detected in 83.3% of patients at baseline. ≥6 CTCs/7.5 mL blood at baseline and post-treatment (cycle 2, day 1) was predictive of worse PFS and OS.
Aggarwal C et al., 2017 [103]	50 (20/30)	CellSearch	CTCs were detected in 94% of patients at baseline. Lower number of CTCs in LD-SCLC (median = 1.5) compared with ED-SCLC (median = 71). Patients with <5 CTCs were associated with longer PFS. Patients with <50 CTCs were associated with longer OS and PFS.
Su Z et al., 2019 [116]	48 (8/40)	CellSearch	CTCs were detected in 100% of patients at baseline. The CNAs analyses predict the response to first line of chemotherapy.
Obermayr E et al., 2019 [112]	48 (4/31/13 unknown)	Parsortix	CTCs were detected in 31.4% of patients at baseline. CTCs in SCLC patients can be assessed using epithelial and neuroendocrine cell lineage markers at the molecular level.
Mohan S et al., 2020 [73]	48 (28/20)	CellSearch	CTCs were detected in 77.1% of patients at baseline 77.1% (24 of the 28 with ED-SCLC and 13 of the 20 with LD-SCLC). The mean and median CTC numbers per 7.5 mL of blood were 523 and 5, respectively (range 0–15,352).
Wang P et al., 2020 [114]	138	Fluorescence in situ hybridization	CTCs could be a predictive and prognostic factor for SCLC. Authors reported a nomogram model using CTCs and a clinical parameter to predict the median survival time of SCLC patients.

Abbreviations: SCLC: small cell lung cancer; CTC: circulating tumor cell; LD-SCLC: limited disease SCLC; ED-SCLC: extensive disease SCLC.; OS: overall survival; CNAs: copy number alterations; PFS: progression-free survival.

On the other hand, single-cell sequencing of CTCs can provide relevant information about the mutational atlas of SCLC. It has been reported that CNAs analyses of single CTCs showed the potential to distinguish SCLC from lung adenocarcinoma [117]. Another study that analyzed CTCs at the single-cell level in a cohort of 31 SCLC patients generated a CNA-based classification to distinguish chemosensitive from chemorefractory disease [11]. More recently, a single-CTC-based CNA score was developed to predict the response to first-line chemotherapy, which provided insight into biomarker development and constituted a convenient approach for clinical disease differentiation [116]. In addition to molecular characterization, CTC isolation in SCLC has allowed the generation of xenograft models and short-term ex vivo cultures for phenotypic and transcriptomic characterization and preclinical drug screening. Thus, circulating tumor cells-derived explants (CDXs) hold promise as additional tools to gain mechanistic insights into treatment sensitivity and resistance [118].

### 3.3. Extracellular Vesicles

The last of the three important circulating biomarkers are circulating extracellular vesicles (EVs), representing a promising option in the oncology field. EVs are a heterogeneous population of cell-derived membranous structures that include exosomes (ranging from 30 to 100 nm), microvesicles (ranging from 50 to 5000 nm) and apoptotic bodies (typically 500–4000 nm) that are released by the cells through various mechanisms [119,120]. They are implicated in cancer progression, favoring the interaction between tumoral and stromal cells, promoting cell proliferation, invasion and distant colonization [121,122,123]. Of note, EVs can be isolated from several body fluids, and patients with cancer normally show higher levels than healthy people [123]. In addition to their increase in circulation as a consequence of the disease, the characterization of their membrane components and their molecular cargo (proteins, mRNAs and ncRNAs, and single- or double-stranded DNA) constitutes a valuable source of biomarkers [119].

#### 3.3.1. Methods to Isolate and to Characterize EVs

Several methods have been reported in order to obtain a better knowledge of EVs and to understand their molecular biology and possible therapeutic value. First, EV isolation can be mainly addressed by ultracentrifugation, immunoaffinity or precipitation strategies, although standardization of methodologies remains a challenge [124]. Ultracentrifugation is the most common isolation method; however, a high amount of start samples are needed and a low yield or insufficient purity of obtained EVs has been reported [125]. For small volumes and specially for exosomes, immunoaffinity capture is often used, obtaining a high-quality yield. Nevertheless, the method is based on the interactions with a membrane antigen; therefore, only EVs positives for a specific antigen are isolated. Specific EVs markers are still the object of investigations. In the last year, several commercial kits based on precipitation of EVs using an agglutinating agent have been developed [125], but again a lack of standardization remains a challenge.

Next, several methods to characterize EVs and to confirm their correct isolation are employed. EVs imaging plays an important role; however, due to the small size of EVs, they often require labeling prior to their subsequent visualization. Electron microscopy, transmission electron microscopy and optical microscopy have been employed to image the different EVs [126]. The Western blot technique is also employed in order to confirm the correct EVs isolation, but is not valid for EVs quantification and a specific EVs housekeeping protein remains unclear [125].

#### 3.3.2. EVs in SCLC

In SCLC, some studies have described the potential of EVs as a clinical tool. For example, Sandfeld-Paulsen et al. found that the CD151 marker was significantly upregulated in the plasma of SCLC after the application of an extracellular vesicle array to phenotype exosomes [127]. Exosomal circulating miR-141 was also upregulated in plasma and serum samples from 122 SCLC patients compared to a control group, and this increase was associated with advanced stages. In addition, the activity of this exosomal miRNA was implicated in angiogenesis, which supported its candidature as a potential novel target for SCLC patients [128]. Another study characterized the miRNA cargo of serum-derived exosomes isolated from 9 SCLC and 11 NSCLC patients and 10 healthy controls. This strategy identified a total of 17 miRNAs differentially expressed in cancer patients. Some of these miRNAs were found to be altered in exosomal samples obtained at baseline and after chemotherapy treatment, evidencing the potential of exosomal miRNAs for developing noninvasive tumor characterization in lung cancer patients [129].

### 3.4. Other Liquid Biopsy Biomarkers

There are other biomarkers in the field of liquid biopsy that are in an emerging state. The expansion of the range of analytes examined might help to obtain more complete information and add value to data obtained from CTCs and cfDNA assessment. MicroRNAs (miRNAs) are small RNA molecules (19–24 nucleotides) that constitute an evolutionary highly conserved class of non-coding RNAs. They play a significant role in the regulation of gene expression, and their discovery has provided new directions for the study of human cancer pathogenesis [130]. Different authors have reported that miRNAs could be useful as tissue biomarkers for classifying lung neuroendocrine neoplasms (NENs) [131,132,133,134,135]. MicroRNAs are also promising tissue biomarkers for examining chemoresistance and can be used as a new therapeutic approach in SCLC [136]. However, circulating miRNAs have been comparatively less studied as blood-based biomarkers in SCLC. Using a 24-miRNA signature previously described for lung cancer in blood [137], metastatic adenocarcinoma and SCLC was predicted with a high accuracy [138]. In the same line, Lu et al. reported two plasma microRNA panels with a considerable clinical value in the diagnosis of lung cancer, being able to perform a relevant role in determining optimal treatment strategies based on discrimination between SCLC and NSCLC [139]. Recently, Li et al. evaluated the expression of candidate circulating miRNAs in SCLC patients to identify potential noninvasive biomarkers [140]. The study revealed that plasma miR-92b and miR-375 might be used as biomarkers of chemotherapy resistance and prognostic factors for SCLC. The presence of circulating miRNAs in plasma and/or serum is highly stable, being detected at low concentrations with great efficiency, so it has huge potential to be utilized as a biomarker in cancer screening and monitoring [141], SCLC in this case. However, there is currently a need for retrospective and prospective larger studies.

The neutrophil-to-lymphocyte ratio (NRL), red cell distribution width (RDW) and platelet-to-lymphocyte ratio (PRL) have been studied as prognostic factors in SCLC patients [142,143]. The NLR is defined as the neutrophil count in blood divided by the lymphocyte count in blood and is a marker for the general immune response to distinct stress stimuli. A correlation between the outcomes of SCLC patients and the NLR has been reported and proposed as a potential biomarker in ES-SCLC patients under different therapies [143] or all stages of SCLC patients under anti-PD-1/PD-L1 therapy [142]. Indeed, elevated RDW and PLR are poor prognostic factors in ES-SCLC patients and LS-SCLC, respectively [143]. However, a recent study did not find an association between these biomarkers and patients receiving anti-PD-1/PD-L1 therapy. Several serum biomarkers have been also reported in SCLC, such as neuron-specific enolase (NSE), lactate dehydrogenase (LDH), serum pro-gastrin-releasing peptide (ProGRP) and carcinoembryonic antigen (CEA) [144].

## 4. Conclusions and Future Perspectives

Liquid biopsy is being incorporated as a useful procedure in many fields, especially in oncology, due to its minimal invasiveness and its great capacity to monitor the disease. The search for new biomarkers will help clinicians manage patients in a personalized way, with the aim of turning the management of cancer into an oncology of precision [145]. The recent development of new technologies has propelled liquid biopsy as a promising diagnostic tool. In recent years, several integrated biological strategies have been used to identify noninvasive markers to predict response and identify new biomarkers in SCLC patients. However, multiple challenges remain before liquid biopsies can be widely used for cancer management, such as the need for standardization of pre-analytical samples and standardized platforms [146].

Recent improvements in cfDNA analysis have made possible the development of cfDNA assays with high sensitivity and specificity, such as WGS, NGS and ddPCR. In SCLC patients, cfDNA-based assays will become increasingly important because they allow minimization of the need for a tumor tissue biopsy. Several studies have found that the levels of cfDNA or CNA are associated with patient outcomes, and good concordance between cfDNA and tissue was reported [71]. In contrast, it is important to remark that the standardized method to isolate and analyze cfDNA is still a challenge. Plasma cfDNA is a complex mixture of DNA from many sources, including germline, fetal, infectious and malignant origins [147], but mutations can also be found on cfDNA of hematopoietic origin, with the risk of considering these mutations as tumor-derived. The incorporation of strict quality controls and/or bioinformatic filtering is critical to ensure that they are specific to the tumor of interest. Therefore, the sequencing of matched PBMCs together with cfDNA would be important for the correct implementation of NGS analyses using plasma samples [72,148].

Analysis of ctDNA could also be potentially integrated in early cancer detection protocols. Lung screening trials showed that lung cancer screening by low-dose computed tomography (LDCT) reduced lung cancer mortality. However, it is necessary to improve the sensitivity and specificity of LDCT and avoid performing invasive tests on patients who do not need them. The combination of imaging and liquid biopsy techniques may be useful in this context. Thus, in the BioMILD assay, it proved the additional value of blood miRNA analysis at the time of LDCT in a large series of volunteers, with the aim of targeting the next LDCT intervals based on an individual risk profile [149].

On the other hand, CTCs are cells with tumor-specific information and can also enable cancer diagnosis and treatment evaluation. In SCLC, the number of CTCs is higher than in other cancer types, including LD-SCLC patients, and this high number of CTCs has been associated with a poor response to different treatments [150]. The high number of CTCs detected in SCLC patients appears to be due to high tumor growth and early disseminative capacity [111]. The CellSearch system has been shown to be the most robust platform to isolate CTCs in SCLC patients, but this platform is only based on the expression of epithelial markers in CTCs; thus, high mesenchymal CTCs can be underestimated using this system. New markers to detect these mesenchymal CTCs would be of great value, together with the validation of new label-independent platforms, which are a promising option to detect the presence of clinical biomarkers, such as PD-L1 or c-MET [151,152]. Another advantage of label-independent platforms is the possibility of performing molecular analyses with isolated CTCs and developing novel therapeutic targets, such as DDL3, the actionable target of Rova-T [112]. Additionally, thanks to methods that allow the preservation and capture of intact CTCs, single-cell analyses or cluster analyses are possible. Single- or cluster-cell characterization of CTCs will allow more precise characterization of the entire cancer in each patient. RNA and DNA analyses are possible and could show the intertumoral and intratumoral heterogeneity in each patient [153]. CDXs models derived from CTCs will also allow for improved knowledge of the mechanisms of tumor development [154]. In contrast, a robust and standardized platform to capture CTCs and rigorous validation remain a challenge.

## Figures and Tables

**Figure 1 biomedicines-09-00048-f001:**
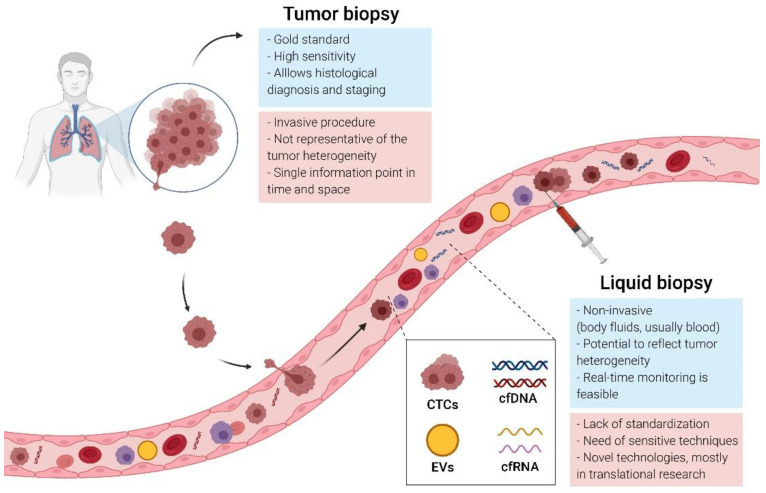
Tissue versus liquid biopsy: comparison of the advantages and limitations.

**Figure 2 biomedicines-09-00048-f002:**
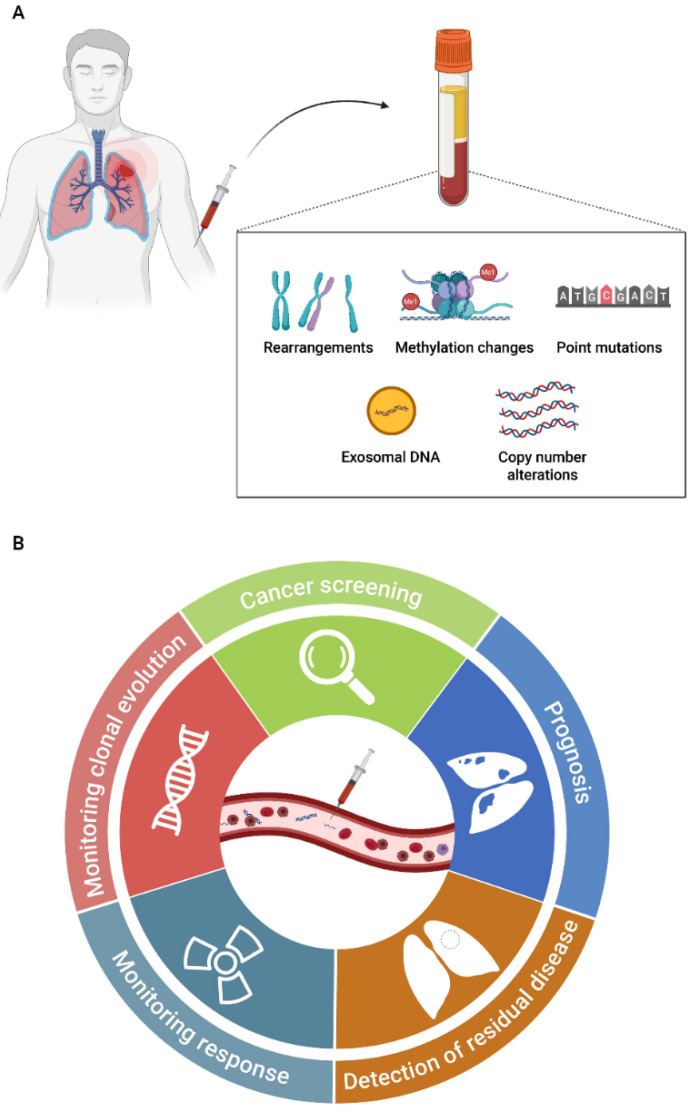
Potential clinical applications of ctDNA in small cell lung cancer (SCLC) patients. (**A**) Range of alterations in cell-free DNA; (**B**) applications of ctDNA analysis during the course of the disease management.

**Figure 3 biomedicines-09-00048-f003:**
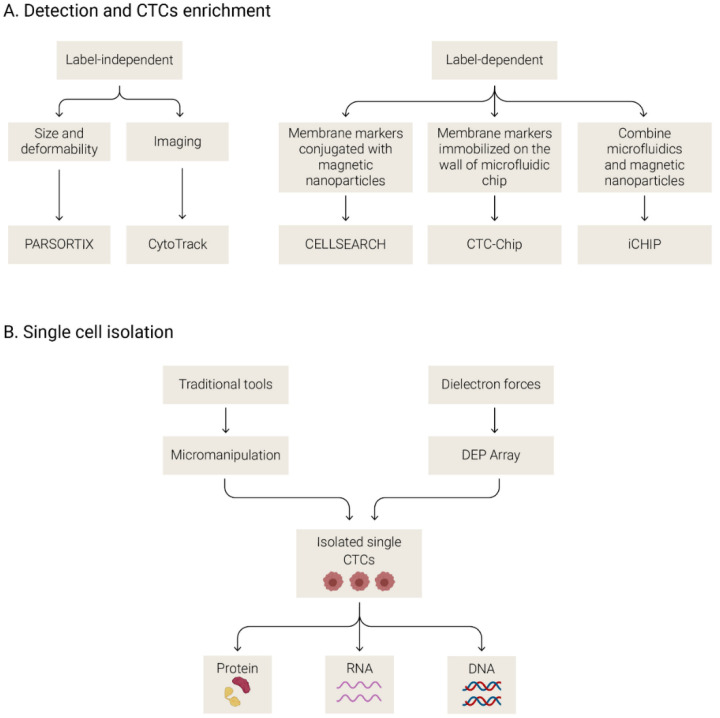
Different strategies for circulating tumor cells enrichment and detection (**A**) and single cell isolation (**B**).

## Data Availability

The data presented in this study are available on request from the corresponding author.

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
