# Peer review of "Current Status and Future Perspectives of Liquid Biopsy in Small Cell Lung Cancer"

_biomedicines, 2021, doi:10.3390/biomedicines9010048_

Round 1

Reviewer 1 Report

  1. Clonal haematopoiesis does not constitute a substantial challenge for the development of ctDNA screening tests since it could be relatively easily addressed by sequencing blood cells, this should be made clear.

  1. Correlation of SLFN11with chemo sensitivity has been described about 8 years ago in https://www.nature.com/articles/nature11003, however how to make use of this correlation in routine clinical settings is still unclear.

  1. CDK4/6 inhibitors is another potential therapeutic approach for subset of SCLCs. About 10% of SCLCs have RB1 WT and subset of such tumors may be sensitive to CDK4/6 inhibitors as described in following 2 articles https://www.ncbi.nlm.nih.gov/pmc/articles/PMC5650296/

, https://www.biorxiv.org/content/10.1101/516351v2

The following clinical trial is investigating use of CDK4/6 inhibitors in RB1 WT SCLC https://clinicaltrials.gov/ct2/show/NCT04010357.

  1. Methylation analysis of ctDNA should be perceived with caution due to effects of age, environmental factors and comorbidities. Heavy smokers are key population for earlier SCLC detection, however smoking by itself can have substantial effect on ctDNA methylation. This caveat needs to be mentioned.

  1. How analysis of ctDNA could be potentially integrated in clinical earlier cancer detection protocols needs to be mentioned, especially keeping in mind use of low dose CT scans in heavy smokers.

Author Response

  1. Clonal haematopoiesis does not constitute a substantial challenge for the development of ctDNA screening tests since it could be relatively easily addressed by sequencing blood cells, this should be made clear.

R: Our apologies for this. The only purpose for our part was to emphasize the idea that paired plasma–peripheral blood cells sequencing should be implemented as the standard practice for NGS genomic analysis of cfDNA to prevent misinterpretation of results. We have rewritten this part of the text, removing the term "challenge". Please see revised manuscript (page7, line 287).

  1. Correlation of SLFN11 with chemo sensitivity has been described about 8 years ago in https://www.nature.com/articles/nature11003, however how to make use of this correlation in routine clinical settings is still unclear.

R: Thank you for the comment. We have included a brief comment about it, referencing a clinical trial (https://clinicaltrials.gov/ct2/show/NCT04334941) (page 4, lines 133-136). 

  1. CDK4/6 inhibitors is another potential therapeutic approach for subset of SCLCs. About 10% of SCLCs have RB1 WT and subset of such tumors may be sensitive to CDK4/6 inhibitors as described in following 2 articles https://www.ncbi.nlm.nih.gov/pmc/articles/PMC5650296/, https://www.biorxiv.org/content/10.1101/516351v2. The following clinical trial is investigating use of CDK4/6 inhibitors in RB1 WT SCLC https://clinicaltrials.gov/ct2/show/NCT04010357.

R: According to the Reviewer’s suggestion, we have included the recommended references and comment (page 3, lines 116-119). 

  1. Methylation analysis of ctDNA should be perceived with caution due to effects of age, environmental factors and comorbidities. Heavy smokers are key population for earlier SCLC detection, however smoking by itself can have substantial effect on ctDNA methylation. This caveat needs to be mentioned.

R: Thank you for the comment. According to the Reviewer’s suggestion, we have mentioned this caveat, including two new references. Please see revised manuscript, page 7, lines 272-275.

  1. How analysis of ctDNA could be potentially integrated in clinical earlier cancer detection protocols needs to be mentioned, especially keeping in mind use of low dose CT scans in heavy smokers.

R: We want to express our appreciation to the reviewer for this suggestion with which we fully agree. For this reason, we have added a brief comment in the Discussion section, including a new reference (ref. 152) (page 16, lines 569-576).

Reviewer 2 Report

Clearly written review

Use of radiation for SCLC not described

Immunotherapy considered standard for ED by some

Minor edits

  1. 4 Figure 1 (type of “high sensitivity” in blue box is cut-off
  2. 5 EV is extracellular vesicle
  3. 5 line 170 replace “poor” with “infrequent”
  4. 5 line 189 add colon after should (should:)
  5. 7 line 279 should ctDNA be cfDNA here?
  6. 9 line 299 include heading “Abbreviations”
  7. 11 line 385 – some discussion on size differences btween CTCs and leukocytes should be added and in particular challenges with applying this approach in SCLC for which CTCs are small in size and similar to leukocytes.
  8. 13 line 431 add “Abbreviations”
  9. 14 line 453 replace “different” with “various”
  10. 15 line 498 “pre-analytical”
  11. 15 line 528 CDX not CDx – define – I don’t think it was defined earlier in the review
  12. 15 line 529 replace “good” with “robust”

Author Response

  1. Use of radiation for SCLC not described

R: Thank you for the comment. According to the Reviewer’s suggestion, we have mentioned the use of radiation therapy in the management of small cell lung cancer, including three new references. Please see revised manuscript (page 2, lines 68-73). 

  1. Immunotherapy considered standard for ED by some

R: We thank the reviewer comment. We have included a brief comment about it (page 2, lines 60-62). We agree with the important role of the immunotherapy in ED-SCLC, and we discuss it in page 3, lines 97-105.

  1. Minor edits

- EV is extracellular vesicle; line 170 replace “poor” with “infrequent”; line 189 add colon after should (should:); line 299 include heading “Abbreviations”; line 431 add “Abbreviations”; line 453 replace “different” with “various”; line 498 “pre-analytical”; line 528 CDX not CDx – define; line 529 replace “good” with “robust”.

R: According to the Reviewer’s suggestions, we have corrected all these issues.

- line 279 should ctDNA be cfDNA here?

R: Thank you for the correction, we have replaced the term ctDNA for cfDNA, being cfDNA a more correct term in this context.

- line 385 – some discussion on size differences between CTCs and leukocytes should be added and in particular challenges with applying this approach in SCLC for which CTCs are small in size and similar to leukocytes.

R: We have included a brief discussion in the context of the methodology to isolate and analyze CTCs (page 11, lines 409-411).

Reviewer 3 Report

The authors reviewed the role of liquid biopsies in small cell lung cancer (SCLC), focusing mainly on cell-free DNA and circulating tumor cells. I think that authors properly organized the manuscript, the paragraphs follow a logical order and, thus, the subject of this review is clear. Nevertheless, I would like to expose major points that should be addressed:

  • English language should be revised throughout the whole manuscript. The proper organization of the paragraphs is penalized by the difficult readability due to the English language.
  • In figure 1 the terms ctDNA and ctRNA do not correlate with the text (page 5, line 166), where cfDNA and cfRNA are mentioned.
  • The short paragraph 3.2.1 could be put together with paragraph 3.2, as they both refers to CTCs characteristics
  • This review could be surely improved by further examining EVs, which have been quickly discussed in paragraph 3.3. Why authors did not discuss the techniques of EVs isolation and their role in SCLC in two different paragraphs, as they did for cfDNA and CTCs? And why authors did not include circulating RNA and miRNA? Considering the title chosen for this review, it should have included all the potential biomarkers found in liquid biopsies for SCLC. The discussion of such aspects would provide a more exhaustive review of liquid biopsies in SCLC.

Author Response

  1. English language should be revised throughout the whole manuscript. The proper organization of the paragraphs is penalized by the difficult readability due to the English language.

R: Our apologies for this. We deeply regret this problem, because we requested assistance to "American Journal Experts" (https://www.aje.com/). Attached Editing Certificate (word version). In any case, manuscript have been rechecked by a native English-speaking.

  1. In figure 1 the terms ctDNA and ctRNA do not correlate with the text (page 5, line 166), where cfDNA and cfRNA are mentioned.

R: Thank you for the correction, this has been amended.

  1. The short paragraph 3.2.1 could be put together with paragraph 3.2, as they both refers to CTCs characteristics

R: According to the reviewer, we have put together both paragraphs.

  1. This review could be surely improved by further examining EVs, which have been quickly discussed in paragraph 3.3. Why authors did not discuss the techniques of EVs isolation and their role in SCLC in two different paragraphs, as they did for cfDNA and CTCs? And why authors did not include circulating RNA and miRNA? Considering the title chosen for this review, it should have included all the potential biomarkers found in liquid biopsies for SCLC. The discussion of such aspects would provide a more exhaustive review of liquid biopsies in SCLC.

R: We really appreciate the comments of the Reviewer about this issue. Other elements, such as circulating miRNA and EVs, had not been included in this review because they are at the early stages of use in SCLC. However, thanks to the comments and suggestion of the reviewer, we have investigated and expanded the part relating to miRNAs and EVs. Please see revised manuscript (Section 3.4. Other liquid biopsy biomarkers: page 15, lines 514-532; and Section 3.3. Extracellular vesicles: pages 14-15, lines 468-473, 480, and 484-498). We hope that the revised version of our manuscript meets all expectations

Round 2

Reviewer 3 Report

Dear authors,

thank you for the revised version of your manuscript. I think that the present form has improved significantly compared to the previous version.